# Novel Biased Normalized Cuts Approach for the Automatic Segmentation of the Conjunctiva

**Giovanni Dimauro** [1,*] and **Lorenzo Simone** [2]

1   Department of Computer Science, University of Bari, 70125 Bari, Italy;
2   Department of Computer Science, University of Pisa, 56127 Pisa, Italy; l.simone3@studenti.unipi.it
*   Correspondence: giovanni.dimauro@uniba.it

**Abstract:** Anemia is a common public health disease diffused worldwide. In many cases it affects the daily lives of patients needing medical assistance and continuous monitoring. Medical literature states empirical evidence of a correlation between conjunctival pallor on physical examinations and its association with anemia diagnosis. Although humans exhibit a natural expertise in pattern recognition and associative skills based on hue properties, the variance of estimates is high, requiring blood sampling even for monitoring. To design automatic systems for the objective evaluation of pallor utilizing digital images of the conjunctiva, it is necessary to obtain reliable automatic segmentation of the eyelid conjunctiva. In this study, we propose a graph partitioning segmentation approach. The semantic segmentation procedure of a diagnostically meaningful region of interest has been proposed for exploiting normalized cuts for perceptual grouping, thereby introducing a bias towards spectrophotometry features of hemoglobin. The reliability of the identification of the region of interest is demonstrated both with standard metrics and by measuring the correlation between the color of the ROI and the hemoglobin level based on 94 samples distributed in relation to age, sex and hemoglobin concentration. The region of interest automatically segmented is suitable for diagnostic procedures based on quantitative hemoglobin estimation of exposed tissues of the conjunctiva.

**Keywords:** semantic segmentation; pattern recognition; hemoglobin; anemia; human tissues; conjunctiva; non-invasive medical device

## 1. Introduction

### 1.1. Background

Anemia is a blood disorder in which the number of red blood cells is inadequate to carry oxygen to human tissues and organs. It affects about a third of the global population, being the most common blood disorder according to the epidemiological results [1–3]. Each different form of this condition has its specific underlying causes. The process of erythrocyte production in the blood involves bone marrow and erythropoietin, a hormone produced by the kidneys, which regulates the process of erythropoiesis, favoring a constant rate of change in the number of erythrocytes in the blood [4]. Adequate production of red blood cells prevents conditions such as anemia and tissue hypoxia. To promote normal erythropoiesis, correct hemoglobin synthesis is required. Hemoglobin, the iron-containing protein, represents the predominant protein found in erythrocytes, responsible for transporting oxygen from the lungs to the other tissues. Anemia caused by deficiencies of the aforementioned factors results in production patterns of abnormal and different erythrocytes [5]. Diagnosing anemia requires in most cases a complete blood count (CBC) to check different properties, including hemoglobin and hematocrit levels. Each physiological need depends on several factors, such as gender, age, different stages of

pregnancy and altitude. The thresholds presented in Table 1 are used to diagnose anemia in individuals in a screening or clinical setting according to World Health Organization diagnostic guidelines [6].

**Table 1.** Hemoglobin (Hb) thresholds used to define anemia living at sea level according to the World Health Organization guidelines [6].

| Age Group | No Anemia | Mild Anemia | Moderate Anemia | Severe Anemia |
| --- | --- | --- | --- | --- |
| Children 5–11 years | $\geq$ 11.5 g/dL | 11–11.4 g/dL | 8–10.9 g/dL | <8 g/dL |
| Children 12–14 years | $\geq$ 12 g/dL | 11–11.9 g/dL | 8–10.9 g/dL | <8 g/dL |
| Non-pregnant women | $\geq$ 12 g/dL | 11–11.9 g/dL | 8–10.9 g/dL | <8 g/dL |
| Pregnant women | $\geq$ 11 g/dL | 10–10.9 g/dL | 7–9.9 g/dL | <7 g/dL |
| Men | $\geq$ 13 g/dL | 11–12.9 g/dL | 8–10.9 g/dL | <8 g/dL |

There has always been a worldwide interest in providing simple, cheap and robust procedures to measure hemoglobin without requiring specialized primary health-care workers or medical laboratories [7]. In response to this need, WHO developed the hemoglobin color scale (HCS) in 2001. It consists of a small card of six shades of red from lighter to darker representing a hemoglobin g/dL concentration from 4 to 14 with a step size of 2 g/dL. The specificity of this method has been disputed in literature; for instance, in 2005 14 studies mostly reported a high sensitivity for detecting anemia (75–97%) [8]. Nevertheless, what is crucial about HCS is its potential for opening the way to different approaches requiring a mixture of expertise from different disciplines, such as computer science, in the future. Like other diagnostic-clinical and analytical-laboratory medical disciplines that are beginning to make extensive use of image, sound or signal analysis; and machine and deep learning techniques [9–18], it is worthwhile to invest in research and development of technologies such as those we deal with in this paper, with the dual purpose of significantly reducing the costs borne by the national health systems and powering the healthcare and medical services that would be exempted from a considerable amount of practically useless activities. Since the importance of the objective evaluation of the pallor of the conjunctiva has been understood, a lot has been done. Numerous researchers have worked to develop methods, techniques and devices to make the estimate of the level of hemoglobin or the determination of the condition of severe anemia, in a non-invasive way, as reliable as possible. We will report a summary of this path in the section "Related Works."

*1.2. Haemoglobin Spectrophotometry*

HCS and physical examination of exposed tissues such as palpebral conjunctiva or nail beds both rely on how humans perceive colors related to the optical spectrum [19]. To better analyze and handle this phenomenon from a computer vision point of view, a chemical insight is required. Spectrophotometry in chemistry is defined as quantitative measurements of the reflective or absorption properties of a material from a wavelength perspective. The spectra of the hemoglobin molecule vary based on whether it is bound to oxygen, carbon monoxide or nothing; the the latter is also called deoxygenated Hb [20].

We relied on experimental literature data [21] for the absorption spectra of hemoglobin used for both plots in Figure 1. The absorption coefficient $\mu_a^{Hb}$ for $HbO_2$ and Hb is calculated as follows:

$$\mu_a^{Hb}(\lambda) = \frac{2.303 \times e_{Hb}(\lambda)\left[\frac{L}{cm \times mol}\right] \times 150[g/L]}{M_{Hb}[g/mol]}, \tag{1}$$

where $e_{Hb}(\lambda)\left[\frac{L}{cm \times mol}\right]$ is the Hb molar extinction coefficient and $M_{Hb}[g/mol]$ is the Hb gram molecular weight, assuming a concentration of 150 grams per liter.

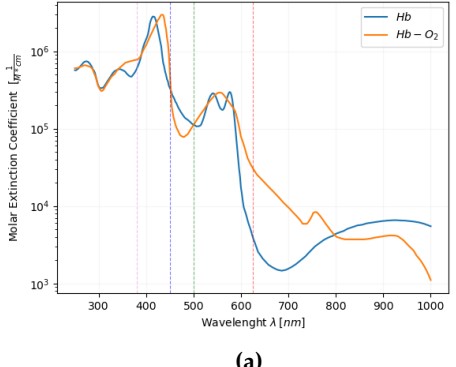 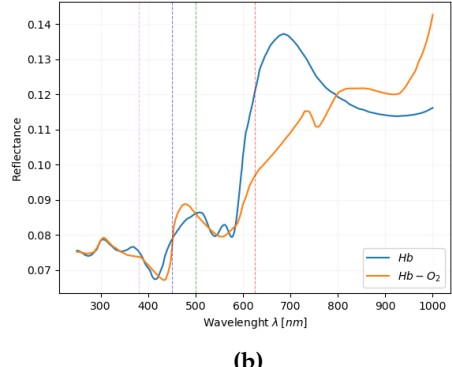

(a)                                                                                (b)

**Figure 1.** Plots visualizing optical absorption and reflectance of Hb and HbO$_2$, vertical dashed lines are related to human perception of colors associated with ($\lambda$). (**a**) Molar extinction coefficient ($\epsilon$) related to absorbance over wavelength ($\lambda$) considering 15 g/dL of hemoglobin concentration and 1 cm cuvette. (**b**) Derived reflectance plot of absorbance under same constants.

Over the years, the palpebral conjunctiva has been a good spot to diagnose anemia, representing a highly vascular area characterized by several capillaries. In [22] a multi-layered tissue model is proposed and investigated to approximate the lower eyelid with seven layers: conjunctival epithelium, tarsal plate, orbicularis oculi, subcutaneous tissue, dermis, epidermis and stratum corneum on the outside of the eyelid tissue. The conjunctiva is perfused from the ascending branch of the posterior conjunctival artery. The presence of interweaving capillary networks penetrating several layers of the model, with the mucous membrane being highly transparent, allows for model approximations for the digital image domain. As already visually described by Figure 1, Hb and HbO$_2$ both absorb wavelengths from 275 to about 550 nm corresponding to a visible spectrum from purple to light green. Each frequency above 600 nm is highly reflected, matching with colors from orange to dark red. A typical human eye is known to be aware of wavelengths in a range from 380 to 740 nm. The cytoplasm of the red blood cell is rich in hemoglobin, that being responsible for the reddish appearance of exposed tissues and blood in general. Laboratory-based experiments conducted in [23,24], inspired us to start from those results to accomplish segmentation and digital image analysis related to hemoglobin.

### 1.3. Related Works

Over the years many researchers have put in effort toward developing non-invasive methods for anemia detection through hemoglobin estimation. The relevance of conjunctiva hue in the clinical evaluation of anemia was tested in [25] for 219 healthy ambulatory subjects. Three educated non-clinicians, appropriately trained, overall agreed on conjunctiva hue performing with kappa coefficients between 0.27 and 0.34. As a result, hue variation strictly depends on the objective of the assessment and training of field personnel. Comparing earlier results obtained by physical examination and the latest digital photography, the latter is minimizing variance, optimizing specificity and sensitivity by using machine learning and automatic segmentation procedures. Establishing the most successful technology still leaves questions about the best region to analyze exploiting color properties associated with better results. Studies in [26] from an ophthalmology point of view open a debate for correlation of anemia between bulbar conjunctival blood column and palpebral conjunctival hue (PCH). From the results of this study, it seems that the bulbar conjunctiva can be successfully included in the set of interesting features, achieving slightly less specificity than PCH, but higher sensitivity. Paradigms of non-invasive and on-demand diagnostics based on smartphone and digital images are spreading due to the advancing of remote diagnosis and affordability [27–29]. A smartphone camera-based application monitoring blood hemoglobin concentration has been developed in [30]. Utilizing a light source pointed to the patient's finger, they performed a chromatic analysis on 31 samples, achieving

sensitivity and precision of 85.7% and 76.5% respectively; they received Food and Drug Administration agreement. Another smartphone-based self-screening tool is depicted in [31] utilizing fingernail beds digital images. Patients select the regions of interest by themselves, corresponding to the nailbeds, and a result is then displayed on the smartphone screen; camera flash reflections and white spots which may affect Hgb level measurements are removed with a quality control algorithm. They reported an accuracy of $\pm 0.92 \text{ g/dL}^{-1}$ of CBC hemoglobin level with personalized calibration, suggesting the relevance of those systems as a monitoring utility. In our study, we analyzed assumptions from related past works and the clinical correlation between conjunctival pallor and anemia condition [32], proposing a fully automated segmentation algorithm. Throughout this process, color features from hemoglobin reflectance spectrum provide a key role in biasing towards a region of interest proposal.

In the literature, few works deal with the automatic segmentation of the conjunctiva. In particular, reference [33] proposes a method for the automatic segmentation of the palpebral conjunctiva that carries out an image processing process based on the equalization of the image in RGB, filter unsharp masking and red channel masking. In [34] the authors developed an algorithm for automatically segmenting the image by finding a "distinctly red" region, bounded by two parallel long-running edges at the top and the bottom; this is achieved by combining the Canny edge detection technique with morphological operations in the CIELAB color space. However, with the aim of estimating anemia, they stated that their method of segmenting was less reliable than manual conjunctiva segmentation made by an expert physician. In [35] the authors use a threshold triangle (which uses triangle algorithm for thresholding) for binary differentiation between the palpebral conjunctiva and background.

### 1.4. Image Capturing Methodology

The technique adopted to capture digital images of a patient's conjunctiva was based on the latest approach of a research study conducted in [36–38]. As a recap, the main requirements to designing an effective tool for estimating the condition of anemia through the use of digital images of the palpebral conjunctiva would be:

- Provide an easy to us;e device with affordable hardware components
- Its usage should not require trained medical personnel;
- It should provide remote diagnosis and telemedicine conveniences.

The acquisition system is shown in Figure 2. It consists of a macro-lens assembled into a specially designed, 3D-printed lightened spacer Figure 2a and a typical smartphone as in the real-life application Figure 2b. The lens can take high-resolution images being attached to a smartphone (we used the Aukey PL-M1 25 mm 10x macro lens). The LED lights can be powered directly from the smartphone or a battery applied to the cover of a smartphone. The lens is fixed on the plastic cover of the smartphone: this device allows for obtaining high resolution images close to the eye, insensitive to the ambient lighting conditions.

The dataset used in the present study, which will be described later, has been created with a Samsung S6 smartphone.

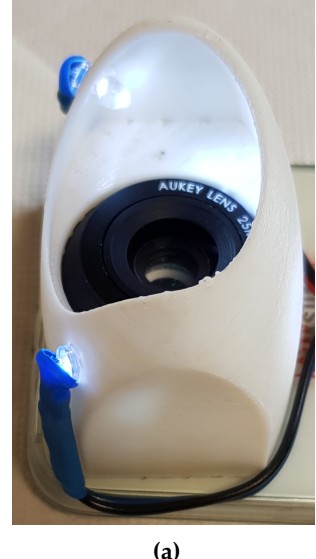

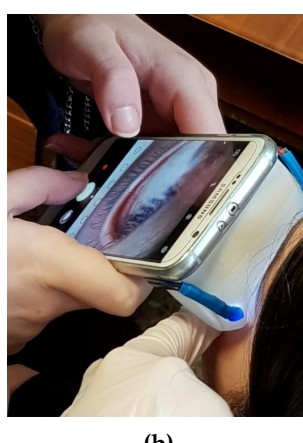

(a)
(b)

**Figure 2.** (**a**) The acquisition device consists of a special spacer and a macro lens to acquire images with a high-resolution smartphone at close range; (**b**) the moment of the acquisition of an image of the conjunctiva.

## 2. Proposed Method

Each digital image from the dataset is converted into an RGB color space matrix representation. The segmentation process can be summarized in three different phases: dimensionality reduction by clustering approach, grouping as graph partitioning and a final ROI extraction. The introduction of a preliminary clustering step determines a speed up in N-Cuts performance arising from the theoretical proofs by the N-Cuts original paper regarding computational complexity in terms of both space and time. The algorithm constructing a region adjacency graph (RAG) does not consider each pixel from the original resolution anymore, but groups of them preserving spatial and color differences amongst them. Finally, we aim at grasping a non-linear relation between brightness intensities from the red and green channels, based on previous assumptions of reflectance rate by a spectrophotometry point of view.

### 2.1. K-Means Dimensionality Reduction

The objective of a clustering task is grouping data instances into subsets maximizing a similarity measure, while different instances should belong to different groups [39–41]. We applied the principles from k-means clustering to image segmentation tasks. The main goal in this phase is to produce a feature space similarly to Voronoi diagrams for planes, reducing the complexity of the graph representing the original image. Each pixel from now on will be referred to as a vector in a five-dimensional space: x and y coordinates from the matrix; R, G and B channel intensities from color representation.

$$f(x,y) = \overrightarrow{p} = \alpha_x \overrightarrow{p_x} + \alpha_y \overrightarrow{p_y} + \alpha_r \overrightarrow{p_r} + \alpha_g \overrightarrow{p_g} + \alpha_b \overrightarrow{p_b} \tag{2}$$

This approach allows us to iteratively minimize the sum of distances from each pixel to its cluster centroid. We briefly summarize the steps of the algorithm as follows:

1. Initialize centroid vectors.
2. Pixels retain spatial as well as color features, allowing us to define an appropriate weighted Euclidean distance as a measure of similarity between them. For each of them, calculate the distance $d$ between the centroid and each pixel of the image defined as:

$$d(\overrightarrow{u}, \overrightarrow{v}) = \| \overrightarrow{u} - \overrightarrow{v} \| = \sqrt{(u_x - v_x)^2 + (u_y - v_y)^2 + (u_r - v_r)^2 + (u_g - v_g)^2 + (u_b - v_b)^2} \quad (3)$$

3. Each pixel is assigned to the centroid minimizing $d$.
4. Recalculate the position of each centroid $c_k$ where $\overrightarrow{p_{ki}}$ is the $i_{th}$ pixel contained in $k_{th}$ centroid using the relation:

$$c_k = \frac{1}{n} \sum_{i=1}^{n} \overrightarrow{p_{ki}} \quad (4)$$

This approach included in the broader field of unsupervised learning approaches, consists of initial batch updates, in which at each step we reassign points to their nearest cluster centroid, followed by cluster centroid recalculations. In online updates, the points are reassigned only if reducing the sum of intra-cluster distances. Those updates already converge towards a local minimum in short order.

In Figure 3, the original image is processed with a three-dimensional (R, G and B) space and in the last picture with a five-dimensional model including both color and spatial features. In the latter, there is not an increase in computational complexity since the only calculation affected is the distance function. However, in each digital image analyzed, the intra-cluster variance is minimized efficiently with properly outlined boundaries in between each group of pixels. The classified instances closer to mucocutaneous junction are noisy in the first approach, while on the second one each semantic class (iris, pupil, sclera, eyelid, and conjunctiva) appears as a compact union of clusters.

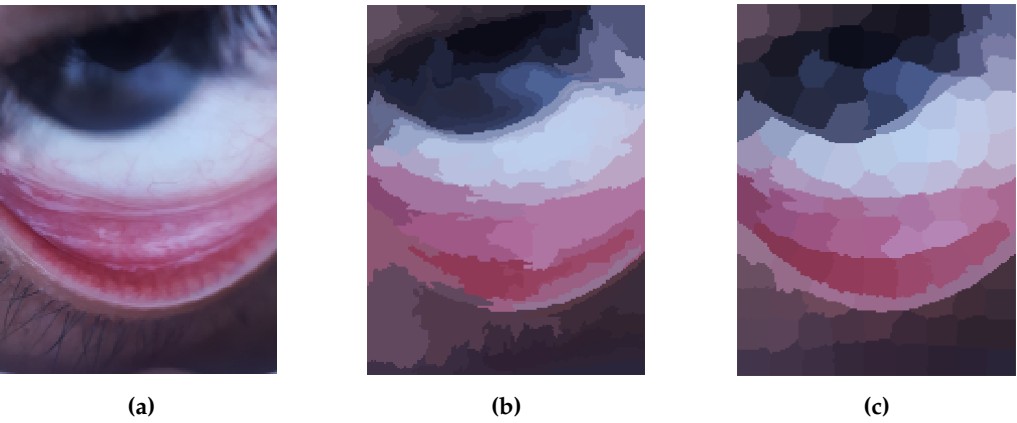

**(a)**　　　　　　　　　　　**(b)**　　　　　　　　　　　**(c)**

**Figure 3.** (**a**) Original digital image acquired; (**b**) k-means clustering procedure using only three dimensional (R,G and B) channels from color space; (**c**) proposed k-means procedure with a model in five dimensions retaining both spatial and color properties.

## 2.2. Normalized Cuts Segmentation

K-means as a clustering algorithm is a valuable approach for exploiting local impressions of a scene, but it lacks in providing a global or hierarchical perspective. For this reason, we take advantage of a grouping algorithm treating the segmentation task as a graph partitioning problem, such as NCuts. It has a better ability to generalize when applied to different scenarios. Conventionally, the normalized cut is an unbiased measure of dissimilarity between graph subgroups [42]. We have converted the set of superpixels from a five-dimensional feature space in a weighted undirected graph $G = (V, E)$. Each point is included in the set of nodes having one edge for each pair of vertices.

The region adjacency graph is constructed based on precomputed areas from the k-means segmentation algorithm. Each connection amongst them is depicted in Figure 4b and representable in a weight matrix W. The edge weight $w_{ij}$ from node $i$ to node $j$ is defined as in the standard approach of normalized cuts as a product of a feature similarity and a spatial term. $X(i)$ is the coordinate vector of the centroid pixel and $F(i)$ is a feature vector based on averaged R, G and B intensities of each pixel in the area. The value $r$ acts as a proximity threshold based on the Euclidean distances amongst precomputed centroids. In our specific application we have tried different configurations ranging from 3 to 100, regulating the sparsity of the weight matrix but not impacting the segmentation outcome. Weights and features are described by the following equations:

$$w_{i,j} = e^{-\frac{\|F(i)-F(j)\|_2^2}{\sigma_I}} * \begin{cases} e^{\frac{-\|X(i)-X(j)\|_2^2}{\sigma_X}}, & \text{if } \|X(i) - X(j)\|_2 < r \\ 0, & \text{otherwise} \end{cases} \tag{5}$$

$$F(i) = \left[ \frac{1}{n}\sum_{j=1}^n p_{jr} \quad \frac{1}{n}\sum_{j=1} np_{jg} \quad \frac{1}{n}\sum_{j=1} np_{jb} \right] \tag{6}$$

The algorithm is capable of extracting significant components from each sample from the dataset, avoiding intra-cluster variations.

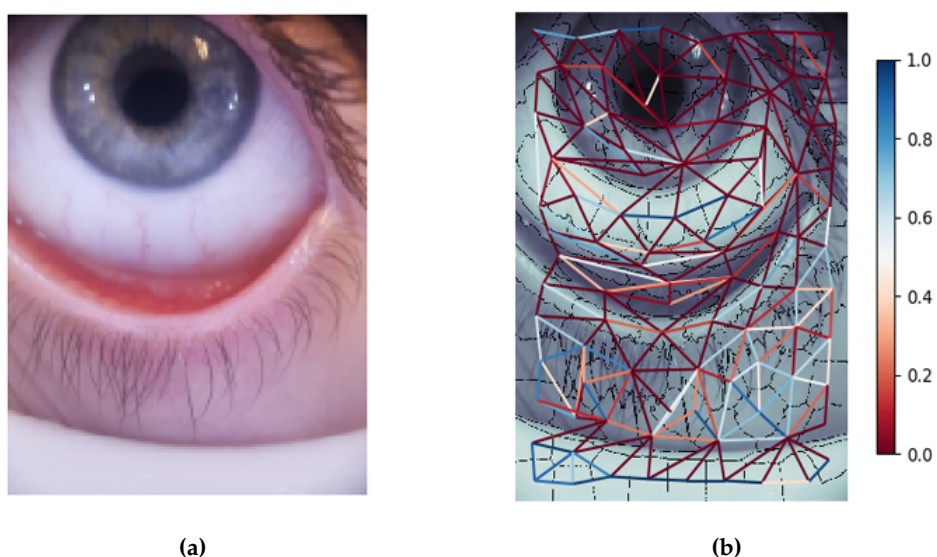

　　　　　　　　**(a)**　　　　　　　　　　　　　　　　　　　　　**(b)**

**Figure 4.** (**a**) Acquired sample; (**b**) region adjacency graph (RAG) displaying a measure of similarity between each region. The center of each node is considered a vertex. For each connection between two regions, there is an associated colored line according to the measure of similarity.

In Figure 5, we added a visual semantic description of the resulting cuts. With this phase, we raise the level of abstraction of the segmentation, starting from the clusters of Figure 3; we end up with features closer to an anatomical perspective. The small gap in colors between the conjunctival area and mucocutaneous junction is perfectly delineated in each sample from the dataset, paving the way for a machine-learning-based anemia estimator.

In the proposed segmentation output from Figure 5, a recursive approach could be run to further decompose regions of interest from the conjunctival area. As an example, this could lead to a better parting of the two conjunctivae, palpebral and forniceal, so as to contribute to the open debate about the prevalence of one or the other as the best estimator of anemia [43]. In fact, the palpebral conjunctiva highlights the vascularization of the underlying area better than the forniceal and probably allows highlighting minimal variations of blood color. The assumption seems confirmed by scientific literature. However, some authors take into consideration the whole conjunctiva, including both palpebral

and forniceal, to construct and validate their models. It is still an open problem. Furthermore, in [43] the authors state that it should be interesting to establish whether the investigations carried out on a small portion of the conjunctiva can be sufficient and position independent. In fact, the sparsity and density of the blood micro-vessels can change in different parts of the eyelid. Therefore, the recursive identification of further clusters can help to answer the above questions.

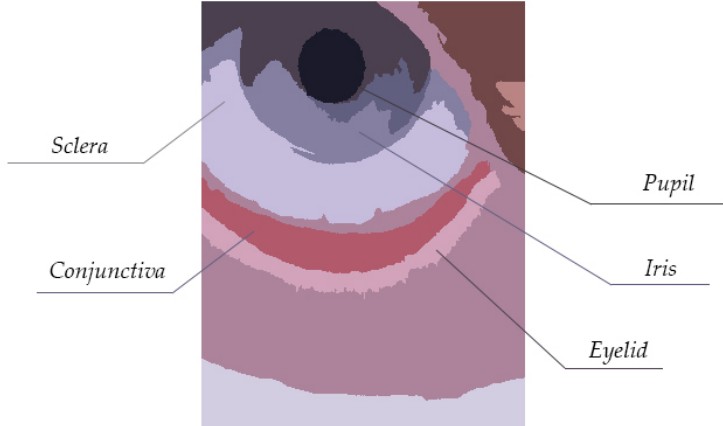

**Figure 5.** Segmentation output result with semantic class description of eye anatomy.

*2.3. Hemoglobin Heatmap Coefficients*

In medical image or radar signal processing tasks, contrast enhancement is a widely used technique in various applications, ranging from improving the quality of photographs acquired in poor conditions [44] to emphasizing regions of interest [45,46]. Histogram equalization is one of the most common approach due to its simple mechanism and effectiveness, but as a drawback, image brightness usually changes after the procedure, caused by its flattening behavior. In our study the objective is focused on approximating the spectrophotometry multi-layered reflectance model investigated in Section 1.2, grasping a mathematical description for digital images. In the literature several studies apply spectral domain scanning, resulting in a time-consuming acquisition process and expensive equipment. This approach does not fit our needs of developing a cheap, non-invasive diagnostic tool. An example of an ill-posed problem known as spectral reconstruction from an RGB scene has been conducted with deep learning techniques in [47,48]. Lastly, researches are highly promoting the validity of these approaches, but despite this, our application domain allows us to further reduce the solution required. Our method, interpreting the image as a signal, performs a pixel pointwise non-linear transformation from red and green color space values, returning a coefficient highlighting vascularized regions. In the literature, the ratio between R and G channels has often been used as a guide to spot those areas, thereby finding the highest values in forniceal and palpebral conjunctival tissues. We propose a generalized logistic function filtering technique including more flexibility than a standard sigmoid. Considering an image I as a vector in three channel functions based on grid coordinates, we obtain the following $\sigma'$ transformation:

$$I(x,y) = \begin{bmatrix} r(x,y) \\ g(x,y) \\ b(x,y) \end{bmatrix}, \qquad \sigma'(I,x,y) = \frac{1}{1+e^{-\alpha\left(\frac{I_r(x,y)}{I_g(x,y)}-\beta\right)}} \qquad (7)$$

The parameter $\alpha$ determines the slope of the function, emphasizing the discrepancy in terms of ratio between color channels; $\beta$ acts as a minimum ratio threshold for the activation of each pixel.

A comparison of the behavior of standard and generalized logistic function with parameterization $\alpha = 4$ and $\beta = 2$ is depicted in Figure 6. This parameterization yielded results with a remarkable

capability of generalizing well in diagnostic imaging ranging from conjunctival tissue to endoscopic domains. Increasing values of $\alpha$ related to the steepness, tend towards the trivial case of a binarization step function losing information about the relationship underlying a variety of brightness ratios. An application of this model is illustrated in Figure 7 useful for digital images of the conjunctival region.

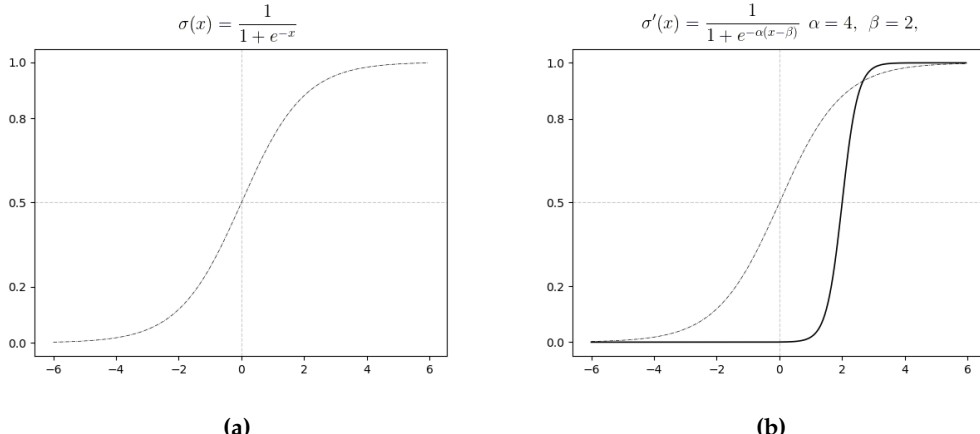

**Figure 6.** (**a**) Standard logistic function plot. (**b**) Generalized logistic function plot using parameters $\alpha = 4$ and $\beta = 2$.

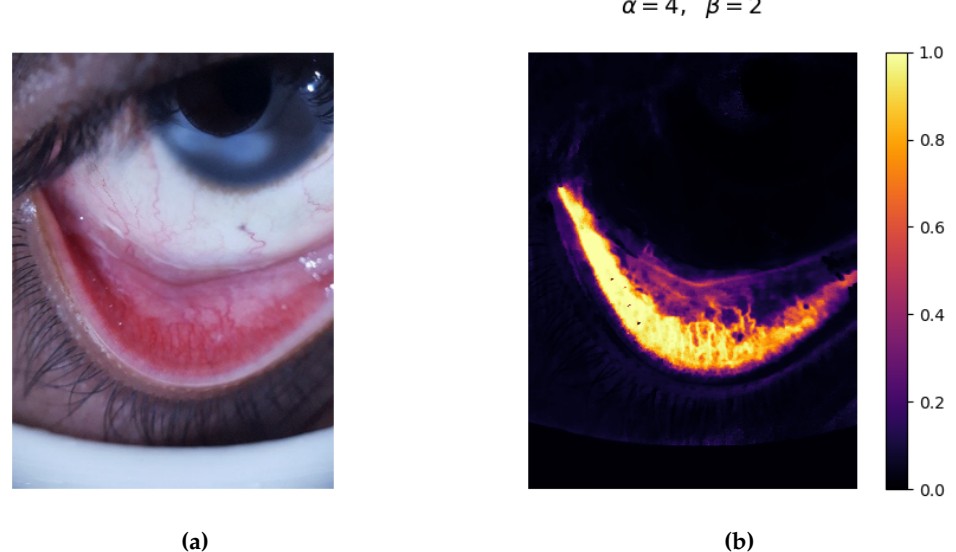

**Figure 7.** (**a**) Acquired sample. (**b**) Heatmap plot of the scoring matrix displaying the magnitudes of the coefficients computed by applying the generalized sigmoid function on the acquired sample.

The real values range from 0 to 1 according to $\sigma'$ function definition. The filtering process produces a scoring matrix assigning lower values to the background, including the sclera, pupil, iris, eyelid and white support platform from the device. Palpebral and forniceal conjunctiva are primarily perfused by both internal and external carotid arteries; this is reflected in high values from the scoring matrix ranging from 0.7 to 1, and the respective blood vessels are significantly highlighted, as shown in Figure 7b.

Since we are interested in obtaining a semantic interpretation out of the regions proposed by NCut, the matrix of coefficients acts as an effective bias for calculating the probability distribution of each class. Edge weights crossed by aggregated pixels resulting from $\sigma'$ are strengthened or decreased, resulting

in a region proposal based on the magnitude of the connection. In Figure 8, we provide a subset of 10 digital images from the dataset, showing the qualitative difference between the proposed semantic segmentation (top row) and the manually segmented ground truth (second row). In Figure 9 we provide two samples of erroneous acquisitions in order to show the robustness of the proposed segmentation in unusual conditions; in fact, only images with excellent characteristics can provide useful information for the correct estimation of anemia.

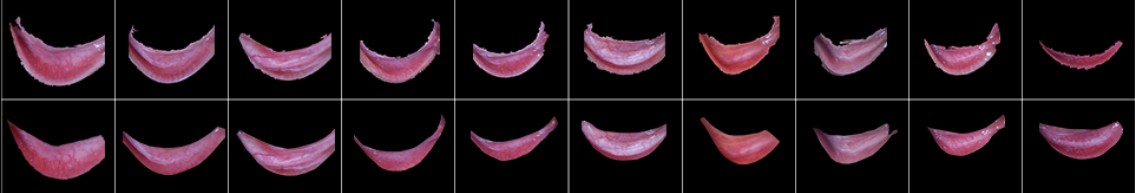

**Figure 8.** The top row represents a subset of samples automatically segmented with the proposed approach. The ordered second row depicts the mapping with the manual segmentation ground truth of the conjunctival region.

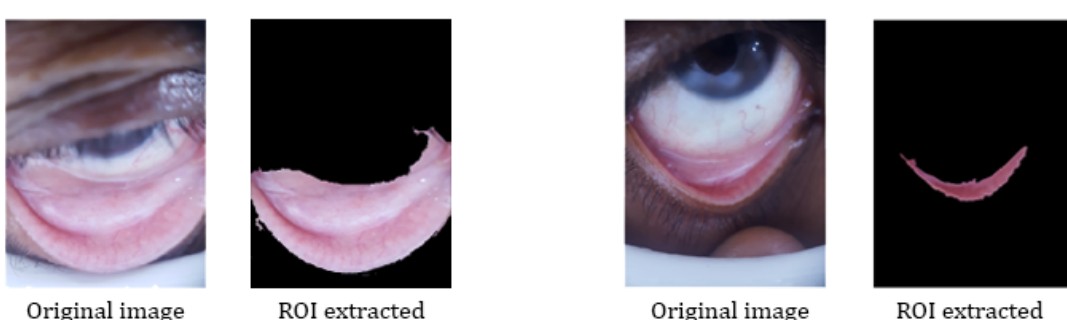

**Figure 9.** Examples of two images that would normally be discarded: the first because the eyelid overlaps the edge of the white spacer and is not perfectly in focus; additionally, the second one is not in focus and the finger appears to lower the eyelid. In both cases, automatic segmentation would still provide an acceptable result.

## 3. Results

The digital images of the patients' eyes have been captured by the device reported in Figure 2 and assembled on a Samsung S6 smartphone; 94 patients were involved, aged 19–75 (average 34), 46 female and 48 male, with Hb level concentrations in the range of 7.6–17.1 g/dL (average of 11.45 g/dL).

Each picture underwent a manual selection process, isolating and cropping regions of palpebral and forniceal conjunctiva, as shown in Figure 10. This step is needed to compare the manually segmented images considered as the ground truth with the automatic segmentation output from the proposed model. We evaluated both spatial and color properties of regions of interest by assessing the most suitable metrics based on this specific medical image segmentation problem [49]. F1 (FMS1), also known as the Sørensen–Dice coefficient, is the harmonic mean of precision and recall, defined as follows for binary segmentation applications:

$$F_1 = 2 \cdot \left( \frac{Precision \cdot Recall}{Precision + Recall} \right) = \frac{2 \cdot TP}{2 \cdot TP + FP + FN} \tag{8}$$

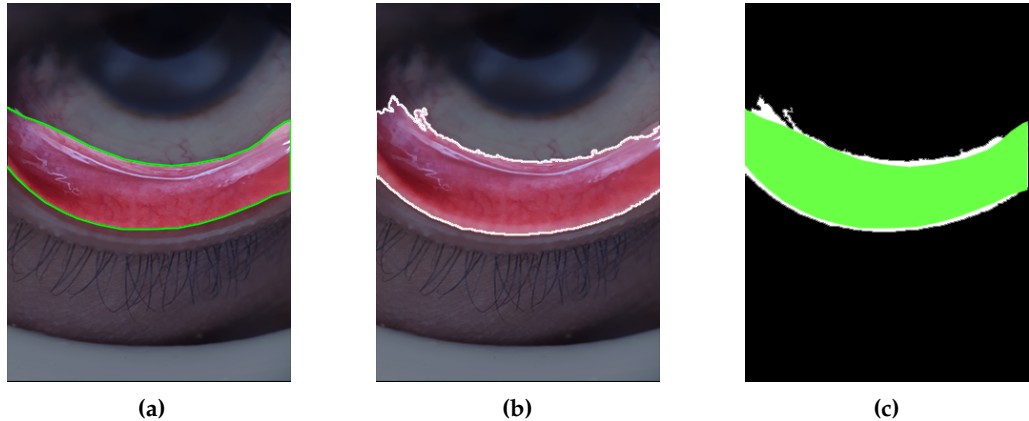

|  | (a) | (b) | (c) |

**Figure 10.** (**a**) Manually segmented conjunctiva used as ground truth. (**b**) Automatically segmented conjunctiva obtained by the proposed approach. (**c**) Visualization of the overlapping between green ground truth image and white automatically segmented image ($F1 = 0.904, accuracy = 96.41\%$).

The Dice coefficient being an overlapping measure ranging from 0 to 1, gives us a useful perspective about the quality of the segmentation. We are also interested in a calculation involving the number of pixels classified as non-relevant (false positive rate), which is not taken into account either by Dice coefficient or by Jaccard similarity. Accuracy metric is helpful in this case by outlining the rate of correctly classified pixels over the full image.

$$Accuracy = \frac{TP + TN}{TP + TN + FP + FN} \tag{9}$$

With the aim of assessing an average for the overlapping metrics, we computed a binary confusion matrix for each image. The values of this matrix refer to the number of pixels linked to set intersection or set difference between ground truth image and proposed segmentation, which are visually described in Figure 10c.

The averaged summation of each confusion matrix is summarized in Table 2. To give the reader the opportunity to observe the indicators for each sample included in the dataset, in Table A1 we have reported the values of the above metrics in a complete manner. Higher values of specificity for this segmentation task highlight the eligibility to disregard non conjunctival regions with proper confidence. On the other hand, sensitivity as well as F1 being overlapping measures, can reasonably fluctuate with higher variance, meaning in most cases that a finer meaningful subset of the conjunctival region has been selected.

**Table 2.** Metrics of averaged results of the comparison between manually and automatically segmented images of the conjunctiva.

|  | F1-Measure | Accuracy | Sensitivity (TPR) | Specificity (TNR) |
|---|---|---|---|---|
| Predicted ROIs | 0.7363 | 93.79% | 86.73% | 94.63% |

The optimal results indicated by the above metrics are sufficient to state the effectiveness of our segmentation algorithm. Since here we are dealing with a rigorous diagnostic procedure, if comparing the precision of the overlapping between proposed and ground truth ROIs is acceptable, we think that a further investigation of the color properties for left-out or added regions would be interesting.

CIELAB is one of the most useful amongst color spaces for erythema analysis and computer vision for diagnostics, composed by an approximately uniform three-dimensional space: L*, a*, b*. A widely used dimension from this space, a*, has a well-known correlation with hemoglobin values in this domain [36–38]. Our purpose is to examine the strength of linear correlation between mean

values of a* extracted from digital images of conjunctivas and the relative Hb g/dL concentration from blood samples taken almost at the same time of picture capturing phase (Figure 11). Generalizing the idea of Pearson correlation coefficient (PCC) from two random variables to two standardized vectors, we can estimate the weight of their linear correlation ranging from −1 to 1 and defined by the following equation:

$$\rho(\boldsymbol{a}, \boldsymbol{b}) = \frac{1}{N-1} \sum_{i=1}^{N} \left( \frac{a_i - \mu_a}{\sigma_a} \right) \cdot \left( \frac{b_i - \mu_b}{\sigma_b} \right) \tag{10}$$

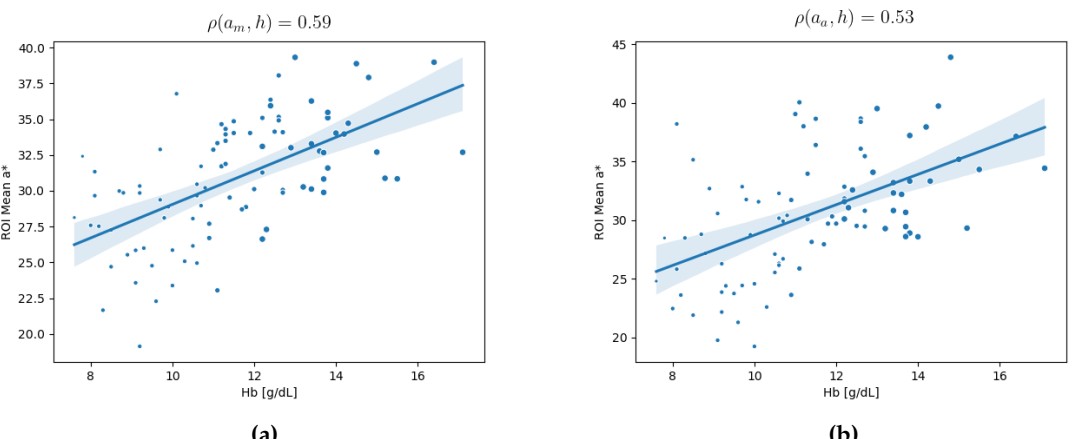

**Figure 11.** (**a**) Linear regression and strength of correlation between a* from manual segmentation and Hb g/dL standardized vectors. (**b**) Linear regression and strength of correlation between a* from automatic segmentation and Hb g/dL standardized vectors.

We computed PCC between the mean a* values for both manually and automatically segmented images and Hb g/dL through the entire dataset of 94 samples, thereby obtaining respectively 0.59 and 0.53. The results reconfirm not only the moderate linear correlation between those values, but also a robust contiguity among human based manual segmentation and fully automated segmentation approach proposed.

## 4. Conclusions

We developed a fully automated segmentation procedure, based on graph partitioning, that exposes conjunctival regions while maximizing the correlation between color properties and hemoglobin concentration in the blood, according to the multi-layered anatomical structures of these tissues. The ROIs extracted by the model underwent an in-depth quantitative comparison with ground truth, using state of the art metrics for similarity and PCC between the a* component from CIELAB space and hemoglobin values. The results attest to the reliability and the capability of generalizing between patients belonging to heterogeneous classes, as the accuracy of the overlap between the manual and automatic ROIs selections, measured with classic metrics, is very good, and the correlation obtained between the level of Hb measured in vivo and that estimated through the color of the manual/automatic ROI are comparable. The proposed method paves the way for further studies involving deep learning techniques for both classifications of an estimated anemia risk category and regression to predict Hb real values. With this study we contribute to the broader diagnostic research field of image processing and analysis of the conjunctival pallor related to anemia diagnosis support. The advancement provided to this non-invasive image capturing procedure will lead to the possibility of embedding the model in a wearable device screening Hb risk category in real-time, without the need for physician support.

**Author Contributions:** The authors contributed equally to this work. All authors have read and agreed to the published version of the manuscript.

**Funding:** This research received no external funding.

**Conflicts of Interest:** The authors declare no conflict of interest.

## Appendix A

**Table A1.** Results computed from the confusion matrices of the comparison between manually and automatically segmented images of the conjunctiva for the entire dataset of 94 samples.

| Image ID | F1-Measure | Accuracy | Sensitivity (TPR) | Specificity (TNR) |
|---|---|---|---|---|
| 164733 | 0.7547 | 0.9097 | 0.6060 | 1.0000 |
| 918410 | 0.7647 | 0.9287 | 0.6369 | 0.9567 |
| 094523 | 0.9123 | 0.9674 | 0.8696 | 0.9595 |
| 103722 | 0.6429 | 0.9687 | 0.6103 | 0.6792 |
| 190841 | 0.7011 | 0.8625 | 0.586 | 0.8724 |
| 154215 | 0.6494 | 0.9558 | 0.5505 | 0.7915 |
| 160737 | 0.7844 | 0.9327 | 0.6470 | 0.9957 |
| 155221 | 0.7179 | 0.9813 | 0.7044 | 0.7319 |
| 122613 | 0.7616 | 0.9176 | 0.8953 | 0.6627 |
| 132714 | 0.6641 | 0.8779 | 0.4971 | 1.0000 |
| 140525 | 0.7250 | 0.9316 | 0.9255 | 0.5959 |
| 154320 | 0.5296 | 0.8965 | 0.3602 | 1.0000 |
| 143315 | 0.7563 | 0.8955 | 0.6081 | 1.0000 |
| 145200 | 0.7834 | 0.9837 | 0.7677 | 0.7997 |
| 150240 | 0.6542 | 0.9170 | 0.4861 | 1.0000 |
| 155237 | 0.7672 | 0.9549 | 0.9374 | 0.6493 |
| 801000 | 0.7595 | 0.9460 | 0.9613 | 0.6277 |
| 121216 | 0.7848 | 0.9521 | 0.6534 | 0.9823 |
| 120556 | 0.6804 | 0.9080 | 0.5207 | 0.9815 |
| 134128 | 0.7827 | 0.9675 | 0.6715 | 0.938 |
| 150536 | 0.8229 | 0.9769 | 0.8237 | 0.8221 |
| 151234 | 0.7343 | 0.9285 | 0.6025 | 0.9400 |
| 155418 | 0.8351 | 0.9757 | 0.8186 | 0.8523 |
| 152136 | 0.7407 | 0.9264 | 0.8862 | 0.6362 |
| 152924 | 0.6875 | 0.9653 | 0.5282 | 0.9846 |
| 153536 | 0.6818 | 0.8958 | 0.5174 | 0.9995 |
| 154129 | 0.8665 | 0.9596 | 0.9719 | 0.7817 |
| 154759 | 0.8436 | 0.9559 | 0.7770 | 0.9226 |
| 155456 | 0.8463 | 0.9539 | 0.8111 | 0.8846 |
| 160045 | 0.6242 | 0.9333 | 0.4544 | 0.9965 |
| 123002 | 0.7943 | 0.9244 | 0.6703 | 0.9745 |
| 122915 | 0.7728 | 0.9664 | 0.7984 | 0.7488 |
| 232040 | 0.6222 | 0.9300 | 0.5065 | 0.8064 |
| 160522 | 0.8019 | 0.9790 | 0.8133 | 0.7909 |
| 121836 | 0.5998 | 0.8646 | 0.5157 | 0.7166 |
| 134745 | 0.7401 | 0.8944 | 0.7800 | 0.7040 |
| 211040 | 0.4881 | 0.9146 | 0.3258 | 0.9724 |
| 210631 | 0.9184 | 0.9838 | 0.9235 | 0.9134 |
| 223744 | 0.7676 | 0.9013 | 0.6468 | 0.9440 |

**Table A1.** *Cont.*

| Image ID | F1-Measure | Accuracy | Sensitivity (TPR) | Specificity (TNR) |
|---|---|---|---|---|
| 224452 | 0.655 | 0.8827 | 0.4872 | 0.9991 |
| 231923 | 0.7167 | 0.9513 | 0.5585 | 0.9999 |
| 232931 | 0.8046 | 0.9636 | 0.7029 | 0.9406 |
| 141804 | 0.7793 | 0.9310 | 0.7248 | 0.8428 |
| 152107 | 0.6693 | 0.9144 | 0.5063 | 0.9871 |
| 161452 | 0.7892 | 0.8955 | 0.7651 | 0.9673 |
| 154641 | 0.8193 | 0.9806 | 0.8627 | 0.7801 |
| 210419 | 0.8587 | 0.9675 | 0.8427 | 0.8753 |
| 221400 | 0.8056 | 0.9256 | 0.6767 | 0.9952 |
| 222325 | 0.8093 | 0.9298 | 0.6913 | 0.9758 |
| 140311 | 0.6237 | 0.9594 | 0.4608 | 0.9645 |
| 180148 | 0.8293 | 0.9154 | 0.7085 | 0.9998 |
| 183506 | 0.7559 | 0.9214 | 0.6226 | 0.9617 |
| 195511 | 0.7103 | 0.9149 | 0.5554 | 0.9849 |
| 201501 | 0.7197 | 0.9031 | 0.5662 | 0.9874 |
| 184029 | 0.7589 | 0.9715 | 0.6305 | 0.9531 |
| 184734 | 0.8508 | 0.9636 | 0.8814 | 0.8221 |
| 185602 | 0.8863 | 0.9722 | 0.8574 | 0.9172 |
| 190638 | 0.8229 | 0.9267 | 0.7120 | 0.9747 |
| 191233 | 0.8163 | 0.9388 | 0.8559 | 0.7801 |
| 191620 | 0.6685 | 0.8737 | 0.7922 | 0.5782 |
| 194457 | 0.7283 | 0.9508 | 0.5858 | 0.9624 |
| 114700 | 0.6357 | 0.9133 | 0.5007 | 0.8705 |
| 115146 | 0.6255 | 0.8800 | 0.5202 | 0.7842 |
| 115853 | 0.8018 | 0.9526 | 0.7490 | 0.8626 |
| 120426 | 0.6434 | 0.9588 | 0.5084 | 0.8762 |
| 202058 | 0.6903 | 0.8737 | 0.5271 | 1.0000 |
| 123714 | 0.7709 | 0.9415 | 0.8038 | 0.7406 |
| 133633 | 0.6015 | 0.9539 | 0.4604 | 0.8673 |
| 143301 | 0.8145 | 0.9803 | 0.7065 | 0.9614 |
| 144551 | 0.7174 | 0.9540 | 0.8865 | 0.6025 |
| 145301 | 0.6573 | 0.9124 | 0.4972 | 0.9693 |
| 150804 | 0.6424 | 0.9447 | 0.4849 | 0.9515 |
| 150539 | 0.8357 | 0.9547 | 0.7311 | 0.9750 |
| 151450 | 0.7388 | 0.9020 | 0.5886 | 0.9917 |
| 153146 | 0.7744 | 0.9295 | 0.6382 | 0.9844 |
| 162916 | 0.7940 | 0.9369 | 0.6713 | 0.9716 |
| 202947 | 0.9040 | 0.9641 | 0.8552 | 0.9587 |
| 180925 | 0.7136 | 0.9124 | 0.6152 | 0.8494 |
| 190130 | 0.8209 | 0.9776 | 0.7666 | 0.8834 |
| 190334 | 0.6594 | 0.9354 | 0.7855 | 0.5682 |
| 121621 | 0.8401 | 0.9549 | 0.7570 | 0.9436 |
| 154729 | 0.4816 | 0.9293 | 0.3244 | 0.9343 |
| 205012 | 0.8539 | 0.9651 | 0.9005 | 0.8120 |
| 205445 | 0.8337 | 0.9887 | 0.8632 | 0.8063 |
| 222551 | 0.7993 | 0.9394 | 0.7278 | 0.8863 |
| 223503 | 0.8563 | 0.9834 | 0.8353 | 0.8783 |

**Table A1.** *Cont.*

| Image ID | F1-Measure | Accuracy | Sensitivity (TPR) | Specificity (TNR) |
|---|---|---|---|---|
| 224240 | 0.7352 | 0.9379 | 0.6334 | 0.8760 |
| 205917 | 0.7118 | 0.9691 | 0.6264 | 0.8242 |
| 225922 | 0.7938 | 0.9492 | 0.8498 | 0.7447 |
| 231050 | 0.7480 | 0.9386 | 0.7003 | 0.8027 |
| 183626 | 0.5987 | 0.9463 | 0.4453 | 0.9133 |
| 161347 | 0.7855 | 0.9466 | 0.7371 | 0.8406 |
| 130148 | 0.6814 | 0.9690 | 0.5243 | 0.9728 |
| 130225 | 0.7896 | 0.9383 | 0.6632 | 0.9757 |

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
