# Peer review of "Novel Biased Normalized Cuts Approach for the Automatic Segmentation of the Conjunctiva"

_electronics, doi:10.3390/electronics9060997_

Round 1

Reviewer 1 Report

1) Please add one or two eyelid images where the eyelid area is not exposed properly, but your algorithm still works on it.

2) The value r acts as a proximity threshold (line 181)

Could you explain how the value of r is calculated?

3) The introduction of a preliminary clustering step determines a speed up in N-Cuts performance (line 139).

Is it researched by someone or you figured it out?

Author Response

ANSWERS TO REVIEWER 1

1) Please add one or two eyelid images where the eyelid area is not exposed properly, but your algorithm still works on it.

Thanks to the reviewer, the suggestion is certainly excellent.

In fact, the clinical protocol that the domain experts have defined with us establishes that the eyelid images where the eyelid area is not exposed or focused properly must be discarded a priori. This because over the years, both we and other authors in the literature have verified that only images with excellent characteristics can provide useful information for the correct estimation of anemia. These rules provide for the broad exposure of the conjunctiva, independence from ambient light, the acquisition of a close and zoomed image and some further. For this reason, we have also designed and patented a special device (see Figure 2): it adapts to the periorbital area, is equipped with autonomous standard lighting, keeps the camera at a fixed distance and includes a macro lens to obtain the best possible image. So the dataset we are using for the clinical protocol is excellent from this point of view, otherwise it cannot be used for the estimation of anemia.

But we think we can respond to the reviewer's suggestion, so we chose two images that would normally be discarded during the acquisition phase: the first because the eyelid overlaps the edge of the white spacer and is not perfectly in focus; also, the second one is not in focus and appears the finger to lower the eyelid. In both cases, automatic segmentation would still provide an acceptable result. We hope this proposal is satisfactory for the reviewer.

We added figure 9 and a short clarification at the end of section 2.3.

Thanks again.

2) The value r acts as a proximity threshold (line 181). Could you explain how the value of r is calculated?

Thank you to the reviewer, we are pleased to better explain this point.

The constant ‘r’ is involved in the calculation of the spatial proximity term multiplying the feature similarity term. Its role in the proposed approach is defining a threshold based on the Euclidean distance amongst precomputed centroids provided by the clustering procedure. In literature [42] several values (r=3-10) have been tested for applications ranging from grouping spatial point sets to color image segmentation. In our specific application we have tried different configurations of ‘r’ scaled to centroid distances finding the tuning of this parameter not impacting the segmentation outcome.

We have clarified this concept in section 2.2

3) The introduction of a preliminary clustering step determines a speed up in N-Cuts performance (line 139). Is it researched by someone or you figured it out?

Thank you to the reviewer, we are pleased to better explain also this point.

Specifically, this is an assumption that we have made according to the theoretical proofs provided by the N-Cuts original paper [42] regarding computational complexity in terms of both space and time. The time required to converge for a standard eigenvalue problem for all eigenvectors takes iterations, furtherly reduced by Lanczos eigensolver method to where n is the number of pixels and m the number of steps for Lanczos convergence.

For instance, considering digital images with a resolution of 1280x720 undergoing the preliminary clustering procedure, this allows us to reduce the magnitude of vertices in the graph from almost  pixels to  centroids, resulting in the improvements that we mentioned for the N-Cuts convergence.

We have clarified this concept in section 2.

Thanks for your suggestion, we believe they have been very helpful in improving our paper.

Reviewer 2 Report

In this paper, the authors present an automated segmentation procedure of the conjunctiva to correlate the colour properties and the haemoglobin concentration in the blood. According to them, with this study they contribute to the diagnostic research field of image processing and analysis of the conjunctival pallor related to anaemia diagnosis support.

In my opinion the paper is well written, technically sound and has merit. Apart from the small minor revisions that I consider (please see below), the reason behind my "Major Revision" decision is that the authors should clearly state what is the difference and improvement with respect to their previous published papers (33 to 35 in the reference list). Only the existance of improvements/major differences between this work under consideration and the previous ones would justify a new and original paper.

As for the minor comments:

  • In the Abstract, I suggest a fluent writting instead of dividing it into different sections.
  • In the text, authors refer to ref. 31 saying that "However, with the aim to estimate anaemia, they stated that their method of segmenting was less reliable than manual conjunctiva segmentation made by an expert
    physician". Have you overcome with your work this? 
  • When you talk about "Paradigms of non-invasive and on-demand diagnostics based on smartphone and digital images are spreading due to the advancing of remote diagnosis and affordability." you could cite some other recent works such as:
    • S. Kanchi, M. I. Sabela, P. S. Mdluli, Inamuddin and K. Bisetty, "Smartphone based bioanalytical and diagnosis applications: A review", Biosensors Bioelectron., vol. 102, pp. 136-149, Apr. 2018
    • P. Escobedo et al., "Smartphone-Based Diagnosis of Parasitic Infections With Colorimetric Assays in Centrifuge Tubes," in IEEE Access, vol. 7, pp. 185677-185686, 2019, doi: 10.1109/ACCESS.2019.2961230.
    • Ogirala, T., Eapen, A., Salvante, K.G. et al. Smartphone-based colorimetric ELISA implementation for determination of women’s reproductive steroid hormone profiles. Med Biol Eng Comput 551735–1741 (2017). https://doi.org/10.1007/s11517-016-1605-7

Author Response

ANSWERS TO REVIEWER 2

In this paper, the authors present an automated segmentation procedure of the conjunctiva to correlate the colour properties and the haemoglobin concentration in the blood. According to them, with this study they contribute to the diagnostic research field of image processing and analysis of the conjunctival pallor related to anaemia diagnosis support.

In my opinion the paper is well written, technically sound and has merit. Apart from the small minor revisions that I consider (please see below), the reason behind my "Major Revision" decision is that the authors should clearly state what is the difference and improvement with respect to their previous published papers (33 to 35 in the reference list). Only the existance of improvements/major differences between this work under consideration and the previous ones would justify a new and original paper.

We thank the reviewer for his appreciation of our paper and we are pleased to better explain its merit.

The advancement of this new research relies mainly on the process of designing a rigorous and fully automated procedure having scalability and generalization capabilities.

The system described in our previous papers [33,34] does not perform the automatic segmentation of the ROI but uses the SLIC Superpixel algorithm to suggest the user (doctor or patient) a possible ROI. This mode is valuable but requires the user to draw a line (let's call it the ROI skeleton) on the device's touch screen. In those two studies, we were therefore still quite far from a fully automatic segmentation.

Our previous paper [35] describes a study the authors consider an interesting contribution to the improvement of the reliability of the Hb estimate. It aims to show that areas of the conjunctiva with different brightness if differently 'weighted' contributes to increasing the reliability of the estimate.

This result is obviously based on the fact that the excessive increase or decrease in brightness, in the pixels of the image of the conjunctiva, reduces the pure chromatic information.

In that work, it is shown that the reasoned elimination of a certain number of pixels from the ROI during the segmentation step (lightest and darkest pixels) makes it possible to increase the correlation of the chromatic features of the ROI with the value of Hb in vivo, but in essence, it does not provide a result that can be immediately used to design an automatic segmentation system.

As for the minor comments:

In the Abstract, I suggest a fluent writing instead of dividing it into different sections.

Good, it looks better to us too, we fixed it.

In the text, authors refer to ref. 31 saying that "However, with the aim to estimate anaemia, they stated that their method of segmenting was less reliable than manual conjunctiva segmentation made by an expert physician". Have you overcome with your work this?

Thanks to the reviewer for this question. The goal we pursued in this work was to automatically achieve ROI segmentation as closely as possible to manual segmentation by experts, with the ultimate aim of designing a fully automatic system, from image acquisition to anemia estimation.

We consider the result obtained to be valid for the following reasons:

  1. a) the accuracy of the overlap between the two selections, measured with classic metrics, is very good;
  2. b) the correlation obtained between the level of Hb measured in vivo and that estimated through the color of the manual ROIs and the ROIs obtained by segmenting the images with the method presented here is almost identical.

For this reason, we believe that the method presented here actually overcomes the result presented in [31].

In the conclusion section we also added these considerations.

When you talk about "Paradigms of non-invasive and on-demand diagnostics based on smartphone and digital images are spreading due to the advancing of remote diagnosis and affordability." you could cite some other recent works such as:

  • Kanchi, M. I. Sabela, P. S. Mdluli, Inamuddin and K. Bisetty, "Smartphone based bioanalytical and diagnosis applications: A review", Biosensors Bioelectron., vol. 102, pp. 136-149, Apr. 2018
  • Escobedo et al., "Smartphone-Based Diagnosis of Parasitic Infections With Colorimetric Assays in Centrifuge Tubes," in IEEE Access, vol. 7, pp. 185677-185686, 2019, doi: 10.1109/ACCESS.2019.2961230.
  • Ogirala, T., Eapen, A., Salvante, K.G. et al. Smartphone-based colorimetric ELISA implementation for determination of women’s reproductive steroid hormone profiles. Med Biol Eng Comput 55, 1735–1741 (2017). https://doi.org/10.1007/s11517-016-1605-7

These are very interesting papers; we certainly add them to the reference section. Thanks!

Thanks for your suggestion, we believe they have been very helpful in improving our paper.

Round 2

Reviewer 1 Report

No comments

Reviewer 2 Report

The authors have addressed my comments and I recommend the publication of this work.